# Group psychosocial interventions for anxiety, depression, and post-traumatic stress disorder in children and adolescents in low- and middle-income countries: A realist systematic review and meta-analysis of randomised controlled trials

Tahir Jokinen[1,2], Mujahed Abassi[3], John Hodsoll[4], Katie H. Atmore[1], Chris Bonell[5], Kelly Rose-Clarke[6]*

**1** Department of Global Health and Social Medicine, King's College London, London, United Kingdom, **2** South London and Maudsley NHS Foundation Trust, London, United Kingdom, **3** Region Skåne, Vårdcentralen Södervärn, Malmö, Sweden, **4** Department of Biostatistics and Health Informatics, Institute of Psychiatry Psychology and Neuroscience, King's College London, United Kingdom, **5** Department of Public Health, Environments and Society, London School of Hygiene and Tropical Medicine, London, United Kingdom, **6** Institute for Global Health, University College London, London, United Kingdom

* kelly.rose-clarke@ucl.ac.uk

## Abstract

Group psychosocial interventions can be a scalable treatment for depression, anxiety, and post-traumatic stress disorder (PTSD) in low- and middle-income countries (LMICs) but effects are heterogeneous. Little is known about how intervention mechanisms and context interact to generate different outcomes in different populations. We conducted a realist systematic review, combining traditional systematic review methods with analysis of intervention mechanisms and contextual factors to further understanding of complex interventions. This involved: (i) a scoping review to build initial theory and inform data extraction and analysis; (ii) systematically searching six databases for randomised controlled trials (RCTs) of group psychosocial interventions for participants aged 0–19 years in LMICs (21 November 2022, updated 26 April 2024); (iii) extracting data on outcomes, contextual factors, and intervention mechanisms to build hypotheses about how context interacts with mechanisms to generate outcomes (context-mechanism-outcome configurations; CMOCs); (iv) narratively synthesising CMOCs into wider programme theories about how and why psychosocial interventions work; (v) meta-analyses and meta-regressions to assess trends and test CMOCs. We included 38 RCTs with data for 6,086 participants (52% female, mean age 13). These data informed 14 programme theories including theories that interventions work best when adapted for participant cognitive level, incorporating age-appropriate activities, accounting for local gender-specific issues, and being culturally adapted. Pooled post-intervention effect sizes were -0.72 (95% CI -1.01 to

**Data availability statement:** The datasets used and/or analysed during the current study are available in Supporting Information.

**Funding:** TJ is funded by an NIHR Academic Clinical Fellowship (ACF-2021-17-009). KR-C is funded by a UKRI Future Leaders Fellowship (MR/W00285X/1). The funders had no role in study design, data collection and analysis, decision to publish, or preparation of the manuscript.

**Competing interests:** The authors have declared that no competing interests exist.

-0.42, p < 0.001) for depression, -0.90 (95% CI -1.57 to -0.23, p = 0.014) for anxiety, and -0.71 (95% CI -1.05 to -0.38, p < 0.001) for PTSD. The only significant results in meta-regressions were larger effect sizes for older children and in populations without exposure to conflict for depression symptoms only. Socio-demographic and contextual factors may influence how interventions work and help to explain the heterogeneity of effects. More high-quality RCTs with moderation and mediation analyses are needed to explore the transferability of these interventions.PROSPERO registration: CRD42022364043.

## Introduction

Child and adolescent mental health is a growing global health priority. The World Health Organisation (WHO) estimates that one in seven adolescents experiences a mental disorder at any given time, accounting for 13% of the global burden of disease [1], and there is likely to be a greater burden of subsyndromal distress that never presents to services. Mental health problems in childhood and adolescence have been associated with issues including poor school achievement and substance misuse [2]. Many mental health problems persist or worsen in later life, leading to significant morbidity and mortality with associated economic burdens [3].

Globally, depression is the fourth highest cause of disability-adjusted life years lost among 10- to 24-year olds. Anxiety disorders are the most prevalent mental disorders among adolescents, affecting 3.6% of 10- to 14-year-olds and 4.6% of 15- to 19-year-olds [4]. Many children and adolescents are exposed to trauma and go on to develop symptoms of post-traumatic stress disorder (PTSD). A systematic review estimated rates of up to 87% for symptoms of PTSD among children and adolescents in low- and middle-income countries (LMICs) exposed to traumatic experiences [5].

Most of the world's children and adolescents live in LMICs. In these settings, the WHO recommends psychosocial interventions as first-line treatment for common mental disorders due to their relative safety compared to pharmacological treatments, accessibility and efficacy [6]. Group interventions are useful because multiple adolescents can be treated at the same time by one facilitator with knock-on cost savings, and may offer additional advantages over individual therapy in terms of the group climate, cohesion and alliance. Group interventions may reduce stigma, potentially allowing participants to speak more openly about problems and receive support from others [7]. Despite WHO recommendations, a recent umbrella review of psychosocial interventions for mental health conditions in LMICs found evidence of effectiveness for children and adolescents to be lacking and inconsistent [8]. A meta-analysis of psychotherapy interventions for youth in LMICs highlighted potential benefits, particularly relative to interventions in high-income countries, but inconsistent effects and significant risk of bias [9]. Developing and testing more effective and scalable interventions is therefore a priority, but a major challenge is the lack of understanding about how interventions work, and the contextual and individual factors that affect

outcomes [10]. Knowing what works, for whom, and in what settings is crucial for policymakers to make informed decisions for their local context.

We aimed to address this research gap using a 'realist systematic review' approach combining realist synthesis with systematic searching and theory-led statistical testing [11]. This modification of the traditional realist review approach [12,13] adds rigour by including all pertinent studies rather than stopping at theoretical saturation, considering study designs in inclusion criteria, and synthesising outcome, moderation and mediation data to determine what light these shed on theorised mechanisms [11]. Key realist terminology is summarised in Table 1.

Our main question was: what are the participant characteristics, contextual factors and intervention mechanisms reported by studies of psychosocial interventions for depression, anxiety and PTSD among children and adolescents in LMICs; and how do they interact to explain differences in effects of such interventions?

Subsidiary questions were:

• How are interventions theorised to work in terms of context-mechanism-outcome configurations (CMOCs)?

• What does qualitative research nested within intervention studies suggest about possible refinements to these CMOCs?

• What do statistical analyses reported by these studies suggest about these CMOCs?

## Methods

We used a modified realist systematic review approach which involved: (i) a scoping review to identify initial theories of interest to focus data extraction and build a theoretical framework to inform subsequent formulation of CMOCs; (ii) a systematic search to include all pertinent studies; (iii) data extraction focused on identifying theory, moderation and mediation analyses, and quantitative outcome data; (iv) a risk of bias assessment; (v) formulating CMOCs and programme theories using reciprocal and refutational synthesis; and (vi) a meta-analysis and meta-regressions of quantitative outcome data to assess overall patterns and test CMOCs and programme theories. Methods are described in more detail below.

We adhered to the Preferred Reporting Items for Systematic reviews and Meta-Analyses (PRISMA) 2020 statement and Realist And MEta-narrative Evidence Syntheses: Evolving Standards (RAMESES) publication standards [15,16]. The review protocol was registered on PROSPERO in October 2022 (CRD42022364043).

**Table 1. Definitions of key realist terminology.**

| | |
|---|---|
| Realist evaluation | A methodology underpinned by the philosophy of critical realism, which assumes that social structures and systems have real effects, and that human actors interact differently with interventions in different circumstances [14]. |
| Context | The setting in which an intervention is implemented, made up of a unique combination of contextual factors. |
| Mechanisms | Causal processes that bring about change within an intervention programme, by triggering changes in the thinking and/or behaviour of participants. |
| Outcome | The observed change resulting from the implementation of an intervention programme. |
| Context-mechanism-outcome configuration (CMOC) | The key analytical unit of realist evaluation; a hypothesis about which aspects of intervention mechanisms are interacting with particular factors in a given context, generating outcomes [14]. |
| Realist review | A theory-driven approach to reviewing the literature that aims to better understand how complex interventions work, for whom, and in what contexts, by combining empirical evidence with theory in order to identify CMOCs. Compared to systematic reviews, pure realist reviews use a purposive and iterative rather than comprehensive and systematic search process to aim for theoretical saturation, and extract data focused on intervention theory rather than quantitative overall outcomes, developing a narrative synthesis [12,13]. |
| Programme theory | An explanatory hypothesis, often built up from multiple CMOCs, that seeks to explain how a given complex intervention operates in particular contexts. |
| Middle-range theory | A more abstract level of theory that offers hypotheses to explain how interventions work across contexts. |
| Grand theory | An abstract overarching hypothesis that seeks to explain the broader functioning of social systems. |

## Scoping review

We conducted the scoping review in September 2022 to identify initial theories of interest to focus data extraction and build a theoretical framework to inform subsequent formulation of CMOCs. We included the five systematic reviews in the umbrella review by Barbui et al [8]. We also included 17 reviews that were excluded from this umbrella review on the basis that they did not conduct a meta-analysis. We included any psychosocial intervention for children and adolescents in LMICs. In the scoping review we included individual or group interventions for any mental disorder. As most studies focused on outcomes for depression, anxiety, and PTSD, we selected these as the outcomes for the systematic search in order to ensure a degree of consistency and generalisability within and across findings. Key findings are summarised here in the Methods section because they informed our approach to data extraction and quantitative analysis in the main review.

We synthesised prospective theories about intervention mechanisms and contextual factors suggested by authors of included studies, and explored to what extent they were supported by findings from related study process evaluations and subgroup/moderation/mediation analyses. Theories could be summarised under four overarching themes requiring further exploration: gender and age [17,18]; social protective factors [19,20]; cultural adaptation [21–23]; and location of intervention delivery [24,25].

## Systematic review

We conducted a systematic review to identify all RCTs of group psychosocial interventions for depression, anxiety, and PTSD in children and adolescents in LMICs in non-clinical settings. Although a pure realist approach would usually involve less stringent restrictions on included study types, we chose to include only RCTs in order to maximise rigour in the quantitative analyses and subsequent conclusions.

## Search strategy

Our inclusion criteria were:

- Population: children and adolescents with a mean age between 0 and 19 years [1].

- Intervention: group psychosocial interventions ('interpersonal or informational activities, techniques, or strategies that target biological, behavioral, cognitive, emotional, interpersonal, social or environmental factors with the aim of reducing symptoms…and improving functioning or well-being') [26].

- Comparator: wait-list control, treatment as usual (TAU), or enhanced treatment as usual (ETAU).

- Outcomes: depression, anxiety, and/or PTSD measured using validated rating scales or locally developed and commonly used alternatives. We did not restrict this to formal diagnostic criteria, but included trials of participants with clinically elevated symptom scores, consistent with other recent meta-analyses [9,27].

- Design: randomised controlled trials (RCTs) conducted in non-clinical settings in LMICs [28].

We did not apply language or date restrictions. We included treatment interventions only and excluded trials of mental health promotion or prevention interventions.

TJ searched MEDLINE, EMBASE, PsycInfo, Global Health (Ovid), Social Policy and Practice (Ovid), and the Cochrane Central Register of Controlled Trials from inception date up to 21 November 2022 (updated 26 April 2024). We combined search terms in title and abstract fields related to children and adolescents, psychosocial interventions, and mental health conditions, and used the Cochrane LMIC filter [29]. Full search strategies are provided in S1 Appendix.

After removal of duplicates, TJ and MA independently screened study titles and abstracts and subsequently full texts, resolving disagreements through discussion. We sought not-yet published results from eligible trials by searching for trial protocols and then contacting authors.

## Data extraction

Using Covidence software TJ extracted information on: study characteristics (country, setting); study design (individual or cluster RCT, number of arms); participants (study population, number per arm, age, gender); intervention (type, theory, location, facilitators, training, exposure, duration, attendance); control condition; outcomes (type, rating scales, mean and standard deviation at baseline/endpoint/follow-up or mean change, rates of follow-up); and theory (insights relating to mechanisms and contextual factors identified by study authors). Data were checked by MA. We contacted study authors to obtain missing data.

## Risk of bias assessment

TJ and MA used the Cochrane Risk of Bias 2 Tool for randomised trials to independently assess risk of bias [30], with disagreements resolved through discussion. No studies were excluded based on risk of bias, but the assessment did inform the synthesis.

## Synthesis of theory

We narratively synthesised theory by building on the themes from the scoping review. We summarised theories of change from included studies, formulating these into CMOCs. We extracted qualitative data including from participant and facilitator interviews and focus groups to identify key concepts, which we used to expand and develop the CMOCs. Where available, we narratively synthesised within-study moderation and mediation analyses and used these to test CMOCs. To raise the level of abstraction, we compared and contrasted CMOCs, noting commonalities shared across similar CMOCs to identify salient features, while highlighting essential differences unique to specific interventions and contexts. We iteratively synthesised these into broader categories of programme theories, organised under the themes identified in the scoping search. We used a reciprocal synthesis method, similar to constant comparison, to examine concepts across studies, CMOCs, and programme theories, and identify where one set of concepts could incorporate another. We used refutational synthesis to explore inconsistencies and contradictions [31].

## Statistical methods

We conducted a meta-analysis of post-intervention standardised mean differences, adjusted for sample size (Hedges' g) to synthesise results by intervention type. We used R version 4.3.2 and the metafor, meta, metasens, and dmetar packages. Given the wide range of settings and treatments, we expected some degree of heterogeneity and therefore used a random effects model. As the summary measure was a standardised mean difference, we used a normal-normal model fit using Restricted Maximum Likelihood (REML). Heterogeneity was quantified with the tau-squared and I-squared statistics. Prediction intervals estimated the expected range of effect sizes (ES) for future studies. For the two studies with two treatment arms we combined the trial of TRT + parenting with trials of TRT alone; [32] and for the other trial we used a multivariate model allowing within-study clustering (assuming correlation to be 0.5).[33] We grouped treatments under five categories: CBT (including CBT, TF-CBT, TRT); IPT; WfR and m-WET; eclectic (EASE, YRI, Shamiri, CBI, BAND, REBT); coping-based interventions (guided self-help, mindfulness, mind-body skills, bibliotherapy, didactic therapy). We conducted several sensitivity analyses; we excluded outliers (trials reporting outcomes with 95% confidence interval (CI) that did not overlap with 95% CI of the pooled effect) and used a Baujat plot to identify studies with high influence on pooled ES according to their contribution to heterogeneity. To investigate aspects of programme theories we tested 6 select study level covariates which could potentially explain the differences in ES across studies: type of intervention (CBT-based, IPT-based, etc.); age of participants (early, middle, and late adolescence); proportion of females in the study; whether study took place in a conflict zone; intervention setting (school, community centre, NGO, or refugee camp); and risk of bias. We used Funnel plots and Egger's regression test as potential indicators of publication or small study bias. Where feasible we

looked at these within type of intervention since heterogeneity in addition to publication bias can lead to funnel plot asymmetries.[34] We used limit meta-analysis to adjust for potential publication bias.[35]

## Results

### Study characteristics

Fig 1 presents the PRISMA diagram. After full-text review, we included 39 reports related to 38 RCTs across 24 LMICs, with data for 6,086 participants. Fifty-two percent of participants were female, with a mean age of 13. Two RCTs targeted children under 10; 27 targeted adolescents aged 10–19; and nine spanned both age ranges. Eight RCTs were feasibility or pilot trials. Table 2 presents study characteristics. The list of excluded studies is available in S1 Text.

Eight RCTs tested group cognitive behavioural therapy (CBT) including its adaptations 'Coping Cat' (n = 2) and Rational Emotive Behaviour Therapy (REBT; n = 1); eleven tested trauma-focused CBT (TF-CBT) including its adaptation 'Teaching Recovery Techniques' (TRT; n = 5). One tested interpersonal therapy (IPT). Six combined CBT and IPT frameworks: 'Early Adolescent Skills for Emotions (EASE; n=3); 'Youth Readiness Intervention' (YRI; n = 1); 'Belonging against Negative Thinking and Depression' (BAND; n = 1); and Shamiri (n = 1). Five tested the eclectic 'Classroom Based Intervention' (CBI). Three tested narrative/written exposure therapy (NET/WET; WfR/m-WET); two mindfulness-based interventions; one didactic therapy; one guided self-help; and one bibliotherapy. Comparators were waitlist (n = 25); TAU (n = 7); and ETAU (n = 7); one RCT had both ETAU and TAU arms [36].

28 RCTs were in schools; four in community centres; five in offices of non-governmental organisations; and one in a refugee camp. Intervention duration ranged from three days to 20 weeks (median seven weeks). Most involved weekly sessions. Intervention facilitators included lay facilitators (n = 15); teachers (n = 3); school counsellors (n = 2); psychologists (n = 18); and psychiatrists (n = 1). Follow-up periods ranged from one week to twelve months.

The most frequent outcome was depression (n = 28); then PTSD (n = 24); and anxiety (n = 13). There was a large amount of diagnostic overlap and comorbidity, with most RCTs addressing two (n = 13) or all three (n = 7) of these outcomes.

### Synthesis of theory: Programme theories

We expanded the initial four themes from the scoping review into five (age, gender, social protective factors and stability, cultural adaptation, and location of intervention delivery), under which we organised programme theories built from CMOCs, summarised in Table 3. We separated the combined theme of age and gender from the scoping review into two separate themes, because these contained very different programme theories.

### Theme 1 - Age

Interventions worked best when content was matched to participants' level of cognitive development (programme theory 1.1), and when they consisted of age-appropriate activities (programme theory 1.2). We reformulated theories of change suggested by authors of included reports as six CMOCs. After analysing evidence from focus groups and interviews, we added an emphasis on participant motivation [33] and adaptation of language [56]. Using a reciprocal synthesis method across CMOCs, we identified from participant, facilitator, and expert feedback that the cognitive developmental level of participants was central to determining the appropriate level of complexity that they could access. This defined programme theory 1.1, which was supported by moderation analyses from two RCTs of an active, more behavioural CBI intervention which found that younger children benefitted more [66,67]. We separated out motivation and engagement, related to fun and interesting age-appropriate activities and less to cognitive complexity, to form programme theory 1.2 [40].

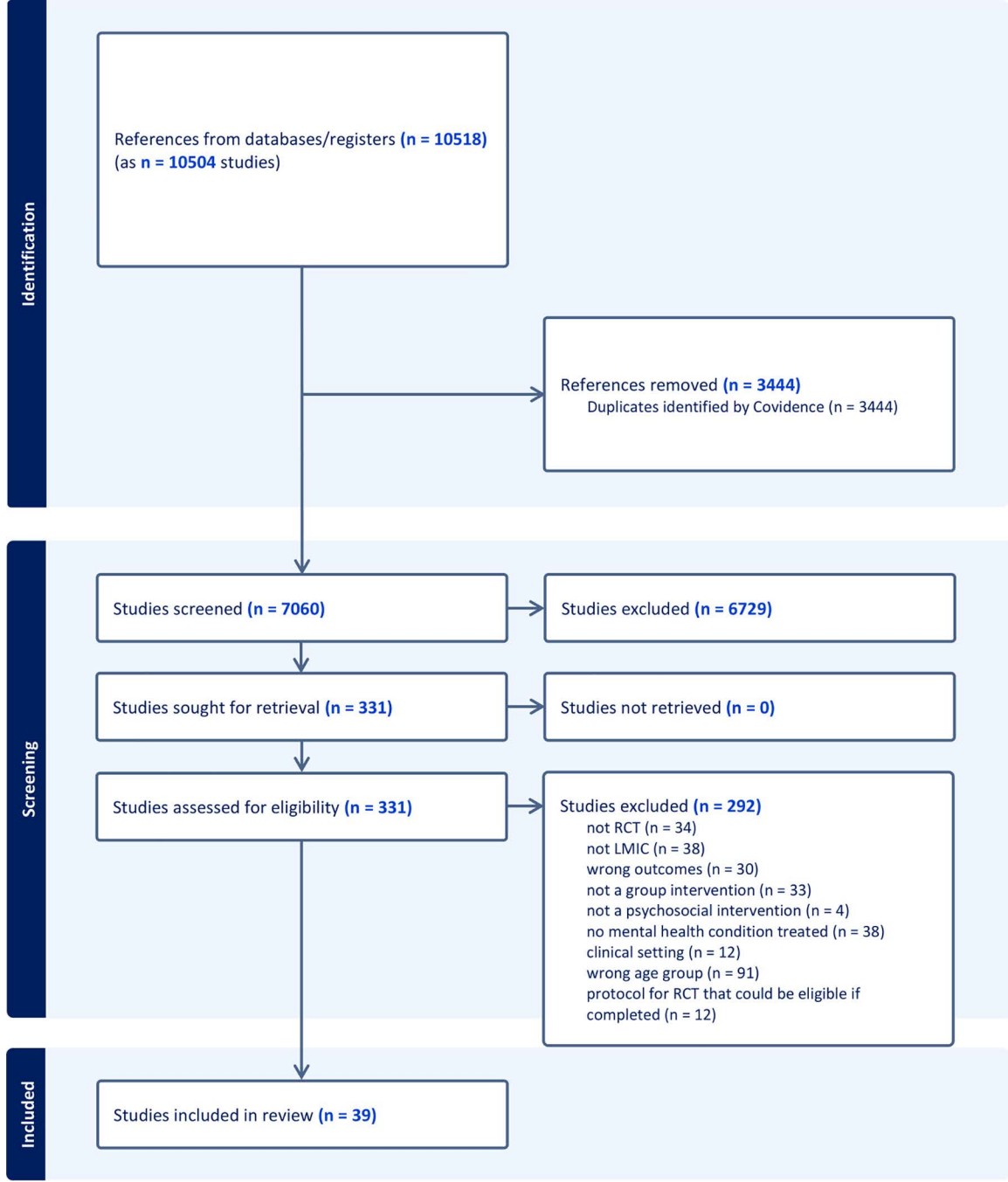

**Fig 1. PRISMA diagram showing search and screening process.**

### Theme 2 - Gender

Gender-specific issues were important contextual factors triggering particular intervention mechanisms (programme theory 2.1). We formulated three CMOCs. Qualitative data from facilitator interviews highlighted the role of aggression and a need for more validation among boys in Brazilian favelas [40]. To refine CMOCs, we distinguished between tendencies

**Table 2. Characteristics of included studies.**

| Study | Country | Study type | Intervention (n participants) | Comparator (n participants) | Setting | Intervention location | Facilitators | Intervention duration | Age range | Participant gender (% female) | Inclusion criteria | Outcomes (scales) | Main results | Subgroup/moderation/mediation analyses | Qualitative evaluation | Theory of change |
|---|---|---|---|---|---|---|---|---|---|---|---|---|---|---|---|---|
| Ahmadi et al., 2022 [33] | Afghanistan | Pilot | m-WET (n=40); TF-CBT (n=40) | Waitlist (n=40) | Urban; Affected by conflict | School | Lay facilitators | 5 days | 12–18 | 100 | Hazara girls attending school exposed to bombing; PTSD score >25 (CRIES-13). | PTSD (CRIES-13) | Significant between-group effects for treatment condition on PTSD severity at post-intervention and three-month follow-up (m-WET and TF-CBT groups had significantly lower PTSD symptom severity than the control group). | N/A | Adolescent and therapist feedback. | Some of the younger girls and those with lower levels of education wrote less because of their writing skill level and m-WET is limited to those who are literate. |
| Akhtar et al., 2021 [37] | Jordan | Pilot | EASE (n=33) | ETAU (n=26) | Urban; Affected by conflict | NGO | Lay facilitators | 7 weeks | 10–14 | 44 | Syrian refugees; psychological distress score >5 (CPDS). | Depression (PHQ-A); PTSD (CRIES-13) | No trend for the remission of distress in either condition. | Mediator: positive parenting (significant interaction effect but no formal mediation analysis). | Semi-structured individual and focus group discussions with key-informants and participants. | The community engagement phase needs to highlight the benefits for the child and family and clarify potential misunderstandings. |
| Barron et al., 2013 [38] | Palestine | Full RCT | TRT (n=90) | Waitlist (n=50) | Urban; Affected by conflict | School | School counsellors | 5 weeks | 11–14 | 45 | Students with the highest PTSD scores (CRIES-13) in each class. | Depression (DSRS); PTSD (CRIES-13) | Significant reductions in PTSD, depression, and secondary outcomes (traumatic grief, negative school impact) in intervention group compared to the waitlist group. | Subgroup analysis: girls had higher levels of traumatic grief throughout. Moderators: student age, counselor gender, school type (no significant effects). | Focus groups with participants and facilitators (thematic analysis). | Students' main learning was 'improved social communication'. Counselors experienced 'working with reputable organizations', 'appreciating new skills', 'a developmentally appropriate program', and 'a collegial context', as motivating. |
| Barron et al., 2016 [39] | Palestine | Full RCT | TRT (n=75) | Waitlist (n=64) | Rural; Affected by conflict | School | School counsellors | 5 weeks | 11–15 | 58 | PTSD score ≥17 on intrusion and avoidance subscales (CRIES-13); depression score ≥12 (MFQ). | Depression (DSRS); PTSD (CRIES-13) | Significant between-group difference only for PTSD but not depression or dissociation. Significant difference for depression in female-only sample. | Moderators: age, school type (not significant); gender (significant for depression only). Mediator: dissociation (not significant). | N/A | TRT appears to provide a protective factor in maintaining symptom levels during ongoing military violence. |
| Barron et al., 2021 [40] | Brazil | Pilot | TRT (n=14) | TAU (n=16) | Urban | NGO | Teachers | 5 weeks | 8–13 | 47 | PTSD score ≥17 on intrusion and avoidance subscales (CRIES-13); depression score ≥12 (MFQ). | Depression (MFQ); PTSD (CRIES-13) | Significant between-group reductions in PTSD and depression, and promising gains in secondary outcome of posttraumatic growth. | N/A | Group interview with facilitators (thematic analysis). | Facilitators had to adapt 'bringing the real world into TRT' because of ongoing crisis in the Favelas. The group environment enabled challenging topics to be raised and discussed, potentially due to the emotionally safe context created and the experiences of healing that occur. |

*(Continued)*

| Study | Country | Study type | Intervention (n participants) | Comparator (n participants) | Setting | Intervention location | Facilitators | Intervention duration | Age range | Participant gender (% female) | Inclusion criteria | Outcomes (scales) | Main results | Subgroup/moderation/mediation analyses | Qualitative evaluation | Theory of change |
|---|---|---|---|---|---|---|---|---|---|---|---|---|---|---|---|---|
| Bella-Awusah et al., 2016 [41] | Nigeria | Full RCT | CBT (n = 20) | Waitlist (n = 20) | Urban | School | Psychiatrist | 5 weeks | 14–17 | 70 | Depression score ≥18 (BDI). | Depression (BDI) | Significant between-group differences in post-treatment depression scores, well maintained at 16 weeks. | Moderators: age, gender (not significant). | Knowledge test. | Behaviourally based interventions are less complicated to explain and easier for patients to understand. Schools offer ready-made venues and social infrastructures that make group-based intervention more readily accessible. |
| Betancourt et al., 2012 [42] | Uganda | Secondary analysis of Bolton et al., 2007 | IPT (n = 103); Creative play (n = 99) | Waitlist (n = 102) | Affected by conflict | Refugee camp | Psychologists | 16 weeks | 14–17 | 53 | Depression score >32 and functional impairment score >0 (APAI); minimum symptom duration one month; residing in displaced-persons camps. | Depression (APAI) | Gender did not, but abduction did significantly predict depression outcomes. In the final model, the three-way interaction was significant. IPT yielded large effect sizes for female subjects without a history of abduction, and male former child soldiers. | Moderators: gender (not significant alone, but significant in interaction with history of abduction), history of abduction (significant). Nonabducted girls and abducted boys had the greatest effect size, abducted girls had some benefit, nonabducted boys had little benefit. | N/A | Former child soldiers experienced stigma on release, and the opportunity for therapeutic support in a social setting may have offset post-conflict difficulties. Former abductees were at risk for continued domestic and community violence and poor economic opportunities, which might compound stigma. Formerly abducted boys appeared to benefit in noticeable ways from the opportunity to discuss their war-related experiences and efforts to reintegrate into life in the IDP camps in the group sessions. |
| Betancourt et al., 2014 [43] | Sierra Leone | Full RCT | YRI (n = 222) | ETAU (n = 214) | Affected by conflict | Community centre | Psychologists | 10 weeks | 15–24 | 46 | Psychological distress score ≥0.5 SD above mean (locally validated measure); self-reported impairment in functioning. | PTSD (PTSD-RI) | No significant difference on PTSD. | Moderators: age, gender (not significant). | N/A | N/A |

*(Continued)*

Table 2. (Continued)

| Study | Country | Study type | Intervention (n participants) | Comparator (n participants) | Setting | Intervention location | Facilitators | Intervention duration | Age range | Participant gender (% female) | Inclusion criteria | Outcomes (scales) | Main results | Subgroup/moderation/mediation analyses | Qualitative evaluation | Theory of change |
|---|---|---|---|---|---|---|---|---|---|---|---|---|---|---|---|---|
| Bolton et al., 2007 [44] | Uganda | Full RCT | IPT (n=105); Creative play (n=105) | Waitlist (n=104) | Affected by conflict; Displaced persons' camps | Refugee camp | Psychologists | 16 weeks | 14–17 | 57 | Depression score >32 and functional impairment score >0 (APAI); minimum symptom duration one month; residing in displaced-persons camps. | Depression (APAI) | Significant between-group difference in depression outcome; creative play not significantly different from waitlist. Among girls, IPT was greatly superior to the waitlist control condition in reducing symptom severity, while the results for boys were not statistically significant. | Moderators: gender (significant); age, abduction history, duration of camp residence (not significant). | N/A | Boys may be less willing to talk about emotional problems, particularly in a group format. The different comorbidity profiles among the boys and girls (boys had more substance use and posttrauma symptoms) may also have affected the effect of IPT. |
| Bryant et al., 2022 [45] | Jordan | Full RCT | EASE (n=185) | ETAU (n=286) | Urban; Affected by conflict | NGO | Lay facilitators | 7 weeks | 10–14 | 50 | Syrian refugees; combined depression/PTSD score ≥15 (PSC-17). | Depression (PHQ-A); PTSD (CRIES-13) | Significant between-group difference at 3-month-follow-up. | Moderator: trauma exposure (not significant). Mediator: reduction in inconsistent disciplinary parenting (significant). | N/A | N/A |
| Chen et al., 2014 [36] | China | Full RCT | CBT (n=16) | ETAU (n=12); TAU (n=12) | Affected by natural disaster | School | Psychologists | 6 weeks | mean 14.50 SD 0.71 | 68 | Adolescents having lost at least one parent in the earthquake; PTSD score ≥18 (CRIES-13). | Depression (CES-D); PTSD (CRIES-13) | Significant between-group differences, especially between the CBT and control groups at the end of treatment and at three-month follow-up. | N/A | N/A | N/A |
| Daryabeigi et al., 2020 [46] | Iran | Full RCT | Coping Cat (n=16) | TAU (n=16) | Rural | School | Psychologists | 8 weeks | 7–10 | 0 | Externalising (T-score >0.63) and internalising (T-score >0.60) disorders (CBCL). | Anxiety (CBCL anxiety/depression subscale); Depression (CBCL withdrawal/depression subscale) | Significant between-group difference in internalising disorders. | N/A | N/A | N/A |
| Dorsey et al., 2021 [47] | Tanzania; Kenya | Full RCT | TF-CBT (n=320) | TAU (n=320) | Mixed | Community centre | Lay facilitators | 12 weeks | 7–13 | 50 | Children having lost one or both parents; residing in the family home; PTSD score ≥18 (CPSS). | PTSD (CPSS) | At post-intervention, TF-CBT was more effective than TAU in urban Kenya, rural Kenya, and urban Tanzania but not in rural Tanzania. At 12-month follow-up, differences by condition were observable only in urban and rural Kenya. In Tanzania, both TF-CBT and TAU groups had substantial reduction. | Subgroup analyses: country versus urban versus rural (significant in urban and rural Kenya, urban Tanzania, not rural Tanzania). More effective in areas with higher stress and adversity. | N/A | N/A |

*(Continued)*

**Table 2.** (Continued)

| Study | Country | Study type | Intervention (n participants) | Comparator (n participants) | Setting | Intervention location | Facilitators | Intervention duration | Age range | Participant gender (% female) | Inclusion criteria | Outcomes (scales) | Main results | Subgroup/moderation/mediation analyses | Qualitative evaluation | Theory of change |
|---|---|---|---|---|---|---|---|---|---|---|---|---|---|---|---|---|
| Ede et al., 2020 [48] | Nigeria | Full RCT | CBT (n = 82) | Waitlist (n = 80) | Urban | Community centre | Psychologists | 12 weeks | 16–25 | 58 | Adolescents with a mobile phone; moderate-severe depression (CES-DC). | Depression (CES-DC) | Significant between-group difference in depression. The mean reduction was maintained after 3 months of treatment. | N/A | N/A | N/A |
| El-Khani et al., 2021 [32] | Lebanon | Full RCT | TRT with parenting skills (n = 41); TRT (n = 38) | Waitlist (n = 40) | Rural; Affected by conflict | NGO | Teachers | 5 weeks | 9–12 | 30 | Score ≥17 on the intrusion or avoidance scales (CRIES-13). | Anxiety (SCARED); Depression (DSRS); PTSD (CRIES-13) | Children in the enhanced group showed significantly greatest levels of improvement on all the total child measures. | N/A | N/A | N/A |
| Getanda et al., 2020 [49] | Kenya | Pilot | WfR (n = 27) | Waitlist (n = 27) | Urban; Affected by conflict | School | Lay facilitators | 3 days | 14–17 | 59 | Experienced at least one traumatic event during the previous year; scoring above the cut-off on any of the emotional problems measures. | Anxiety (RCMAS); Depression (DSRS); PTSD (CRIES-13) | Significant time effect for PTSD, and time x group for PTSD/anxiety/depression. | N/A | Participant free text on experience (content analysis). | N/A |
| Gordon et al., 2008 [50] | Kosovo | Full RCT | Mind-body skills (n = 41) | Waitlist (n = 41) | Urban; Affected by conflict | School | Teachers | 6 weeks | 14–18 | 76 | Score of 3–4 on at least 1 out of 4 reexperiencing symptoms, 3 out of 7 avoidance/numbing symptoms, 2 out of 5 arousal symptoms | PTSD (HTQ) | The intervention group had significantly decreased PTSD scores following the program compared to the control group, representing a large effect size. PTSD scores remained decreased at the 3-month follow-up of the first intervention group. The PTSD score at 3-month follow-up was not significantly different from the PTSD score immediately following the program, and it remained significantly lower than the baseline score. | N/A | N/A | N/A |

*(Continued)*

**Table 2.** (Continued)

| Study | Country | Study type | Intervention (n participants) | Comparator (n participants) | Setting | Intervention location | Facilitators | Intervention duration | Age range | Participant gender (% female) | Inclusion criteria | Outcomes (scales) | Main results | Subgroup/moderation/mediation analyses | Qualitative evaluation | Theory of change |
|---|---|---|---|---|---|---|---|---|---|---|---|---|---|---|---|---|
| Jacob et al., 2016 [51] | Philippines | Full RCT | Bibliotherapy (n=15) | TAU (n=15) | Urban | School | Psychologists | 6 weeks | 13–16 | 100 | Female high school students with high depression scores (BDI-I>14, AADS>61, KADS-11>12) | Depression (BDI) | Participants in the experimental group had experienced a significant alleviation of their depression. | N/A | N/A | When groups are solely composed of females, the intervention produces significant effects because females are more likely to discuss sensitive issues that influence their mood in female-only groups. |
| Jordans et al., 2010 [17] | Nepal | Full RCT | CBI (n=164) | Waitlist (n=161) | Rural | School | Lay facilitators | 5 weeks | 11–14 | 49 | Children screened for generic psychosocial distress | Anxiety (SCARED); Depression (DSRS); PTSD (CPSS) | Moderate effect sizes were found for function impairment, prosocial behaviour, generic psychological difficulties, and depression, and a small effect size for anxiety. | Moderators: gender (significant for girls improving more on prosocial behaviour, boys on psychological difficulties and aggression), age (not significant) | N/A | CBI is especially beneficial for boys by reducing aggression and behaviour-oriented psychological difficulties. This may be explained by the relatively active nature of CBI, which is more compatible with externalising expressions of distress, typical among boys. |
| Jordans et al., 2023 [52] | Lebanon | Full RCT | EASE (n=80) | ETAU (n=118) | Urban; Affected by conflict | NGO | Lay facilitators | 7 weeks | 10–14 | 49 | Adolescents screened for generic psychosocial distress | Depression (PHQ-A); PTSD (CRIES-13) | There were no significant differences between groups. Post hoc sub-group analyses suggest that the control condition may outperform EASE on psychological distress and internalising problems for the subgroup with higher distress; and EASE may outperform the control groups on the same outcomes for the subgroup with lower distress. | N/A | N/A | The lack of superiority of EASE raises the question whether young adolescents, especially when experiencing severe adversity (ongoing economic hardship), can genuinely adopt strategies in a group format and apply them to their own life. |
| Kaesornsamut et al., 2012 [53] | Thailand | Full RCT | BAND (n=30) | Waitlist (n=30) | Urban | School | Psychologists | 7 weeks | 16–18 | 48 | Adolescents with mild to moderate depressive symptoms (CES-D score 16–29) | Depression (CES-D) | Adolescents receiving the BAND intervention, compared with those in the control group, significantly increased their sense of belonging, and decreased their negative thinking and depressive symptoms. | N/A | N/A | N/A |

*(Continued)*

| Study | Country | Study type | Intervention (n participants) | Comparator (n participants) | Setting | Intervention location | Facilitators | Intervention duration | Age range | Participant gender (% female) | Inclusion criteria | Outcomes (scales) | Main results | Subgroup/moderation/mediation analyses | Qualitative evaluation | Theory of change |
|---|---|---|---|---|---|---|---|---|---|---|---|---|---|---|---|---|
| Kalantari et al., 2012 [54] | Iran | Full RCT | WfR (n = 29) | Waitlist (n = 32) | Urban; Affected by conflict | School | Psychologists | 3 days | 12–18 | 48 | War bereaved Afghani students with the highest scores on TGIC | PTSD (TGIC) | The average score in the experimental group decreased significantly, while it slightly increased fin the control group. | N/A | N/A | N/A |
| Khalid et al., 2022 [55] | Pakistan | Pilot | Guided self help (n = 28) | Waitlist (n = 27) | Urban | School | Psychologists | 8 weeks | 13–14 | 46 | Adolescents scoring above the cutoff of 5 on GAD and/or PHQ | Anxiety (GAD-7); Depression (PHQ-9) | There was a significant reduction in symptoms of depression and anxiety in experimental group after receiving the intervention, and this difference in symptoms remained significant from pre-intervention to 3-month follow-up. | N/A | Focus groups with children when adapting the intervention. | N/A |
| Khan et al., 2020 [56] | Pakistan | Pilot | Coping Cat (n = 12) | Waitlist (n = 12) | Mixed | School | Psychologists | 12 weeks | 8–13 | 0 | Boys living in an orphanage having anxiety score >30 (SCARED) | Anxiety (SCARED) | CC program significantly reduced children's self-reported anxiety sensitivity and anxiety symptoms at post-assessment and 6-month follow-up as compared to a waitlist control group. | N/A | Expert panel with psychologists and focus group with children when adapting intervention. | N/A |
| Layne et al., 2008 [57] | Bosnia | Full RCT | CBI (n = 66) | ETAU (n = 61) | Affected by conflict | School | Lay facilitators | 17-20 weeks | 13–19 | 65 | Adolescents with significant trauma exposure before, during, and/or after the war; with severe persisting symptoms of PTSD, depression, or traumatic grief; and significant functional impairment | Depression (DSRS); PTSD (PTSD-RI) | No significant difference between groups. | Moderator: bereavement (significant - more effective) | Evaluation with participants and facilitators. | N/A |

(Continued)

**Table 2.** (Continued)

| Study | Country | Study type | Intervention (n participants) | Comparator (n participants) | Setting | Intervention location | Facilitators | Intervention duration | Age range | Participant gender (% female) | Inclusion criteria | Outcomes (scales) | Main results | Subgroup/moderation/mediation analyses | Qualitative evaluation | Theory of change |
|---|---|---|---|---|---|---|---|---|---|---|---|---|---|---|---|---|
| Li et al., 2022 [58] | China | Pilot | TF-CBT (n=45) | TAU (n=42) | Urban | School | Lay facilitators | 10 weeks | 9–12 | 38 | Children who had experienced at least one traumatic event meeting DSM-5 criteria for PTSD or partial PTSD assessed with the PCL-5 | PTSD (PTSD-RI) | The overall difference between the two groups was not at a statistically significant level. | N/A | Participant feedback. | N/A |
| Li et al., 2023 [59] | China | Full RCT | TF-CBT (n=118) | TAU (n=116) | Urban | School | Lay facilitators | 9 weeks | 9–12 | 41 | Children who had experienced at least one traumatic event meeting DSM-5 criteria for PTSD or partial PTSD assessed with the PCL-5 | Anxiety (SCARED); Depression (CDI-S); PTSD (PTSD-RI) | The intervention group had a significantly larger reduction in PTSD, depression, and anxiety symptoms at post-intervention. At 3-month follow-up, the difference between groups was no longer significant. | N/A | Participant feedback. | N/A |
| McMullen et al., 2013 [22] | Democratic Republic of the Congo | Full RCT | TF-CBT (n=25) | Waitlist (n=25) | Urban; Affected by conflict | School | Psychologists | 15 weeks | 13–17 | 0 | Male former child soldiers (abducted or recruited by an armed group) or witness to a violent event involving a real or perceived direct threat to life; adolescents with suicidal ideation, substance abuse or other mental health difficulties were not excluded | PTSD (PTSD-RI) | Significant reduction in post-traumatic stress, depression/anxiety-like symptoms, conduct problems and increase in prosocial behaviour. Slightly larger effect size for former child soldiers only when excluding other war-affected boys. | Former child soldiers (sub-group analysis - significant, larger effects). | N/A | N/A |
| O'Callaghan et al., 2013 [60] | Democratic Republic of the Congo | Full RCT | TF-CBT (n=24) | Waitlist (n=28) | Urban; Affected by conflict | School | Lay facilitators | 5 weeks | 12–17 | 100 | Girls having witnessed or having personally experienced rape or inappropriate sexual touch | Anxiety/Depression (APAI-AYPA); PTSD (PTSD-RI) | The TF-CBT treatment group had a highly significant reduction in trauma symptoms with a very large effect size, a highly significant reduction in depression and anxiety with a very large effect size. | N/A | Focus groups | N/A |

*(Continued)*

**Table 2.** (Continued)

| Study | Country | Study type | Intervention (n participants) | Comparator (n participants) | Setting | Intervention location | Facilitators | Intervention duration | Age range | Participant gender (% female) | Inclusion criteria | Outcomes (scales) | Main results | Subgroup/moderation/mediation analyses | Qualitative evaluation | Theory of change |
|---|---|---|---|---|---|---|---|---|---|---|---|---|---|---|---|---|
| Osborn et al., 2021 [61] | Kenya | Full RCT | Shamiri (n = 205) | ETAU (n = 208) | Urban | School | Lay facilitators | 4 weeks | 13–18 | 65 | Adolescents self-reporting moderate-severe depression symptoms (≥15 on PHQ-8) or anxiety symptoms (≥10 on GAD-7) | Anxiety (GAD-7); Depression (PHQ-8) | The Shamiri intervention led to greater reductions in symptoms of depression and anxiety post-treatment and extending to 7-month follow-up. | N/A | N/A | N/A |
| Peter et al., 2022 [62] | India | Full RCT | MBCT (n = 36) | Waitlist (n = 36) | Urban | School | Psychologists | 12 weeks | 12–14 | 52 | Adolescents attending school experiencing anxiety more than mild severity based on T-score >0.60 (SCAS) | Anxiety (SCAS) | MBCT aided in reducing anxiety and improving mindfulness traits and resilience but the same effect was not observable in the waitlist group. | N/A | N/A | N/A |
| Pityaratstian et al., 2015 [63] | Thailand | Full RCT | TRT (n = 18) | Waitlist (n = 18) | Affected by natural disaster | School | Psychologists | 3 days | 10–15 | 72 | Adolescents with a primary diagnosis of DSM-IV-TR PTSD | PTSD (PTSD-RI) | Participants assigned to CBT showed greater improvement than those in the waitlist condition. The improvement was not statistically significant at post-treatment, but after a one-month period in which children were encouraged to monitor their symptoms and use relevant techniques, significant improvement was observed. | N/A | N/A | N/A |
| Saw et al., 2019 [64] | Malaysia | Pilot | CBT (n = 10) | Waitlist (n = 10) | Urban | School | Psychologists | 8 weeks | 16 | 50 | Adolescents with a total raw score above the cut-off point of 76 (RADS-2) | Depression (RADS-2) | Compared to a control group, intervention participants showed significant reductions in depressive symptomatology and automatic negative thoughts over the course of the intervention as well as 1-month post-intervention. | N/A | Expert feedback for adaptation; participant focus group and written feedback for intervention. | N/A |

*(Continued)*

**Table 2.** (Continued)

| Study | Country | Study type | Intervention (n participants) | Comparator (n participants) | Setting | Intervention location | Facilitators | Intervention duration | Age range | Participant gender (% female) | Inclusion criteria | Outcomes (scales) | Main results | Subgroup/moderation/mediation analyses | Qualitative evaluation | Theory of change |
|---|---|---|---|---|---|---|---|---|---|---|---|---|---|---|---|---|
| Saw et al., 2020 [65] | Malaysia | Full RCT | CBT (n = 42) | Waitlist (n = 43) | Urban | School | Psychologists | 8 weeks | 16 | 59 | Adolescents with a total raw score above the cut-off point of 76 (RADS-2) | Depression (RADS-2) | CBT intervention corresponded, on average, with a statistically significant reduction in depressive symptoms that continued through 3-month follow-up after-intervention. | N/A | Teacher and counsellor feedback. | Provided that there was no age gap between the participants, the elements and range of discussion during the intervention possibly enhanced the understanding and absorption of the module, especially the cognitive component. |
| Tol et al., 2008 [23] | Indonesia | Full RCT | CBI (n = 182) | Waitlist (n = 221) | Affected by conflict | School | Lay facilitators | 5 weeks | 7–15 | 48 | Symptom checklists assessing exposure to violent events (minimum 1), PTSD (minimum 11), and anxiety complaints (minimum 5) | Anxiety (SCARED); Depression (DSRS); PTSD (CPSS) | Moderate reduction in PTSD symptoms and function impairment for girls and retained hope for boys and girls in comparison to a wait-listed condition between baseline, 1-week, and 6-month follow-up, in a situation of ongoing in-security and instability. | Moderator: gender (significant, girls more effective). | N/A | N/A |
| Tol et al., 2012 [66] | Sri Lanka | Full RCT | CBI (n = 199) | Waitlist (n = 200) | Affected by conflict | School | Lay facilitators | 5 weeks | 9–12 | 39 | Children screened for exposure to traumatic events, post-traumatic stress symptoms, or depressive anxiety symptoms, with the use of symptom checklists | Anxiety (SCARED); Depression (DSRS); PTSD (CPSS) | Intervention effects were identified for children experiencing lower levels of current war-related daily stressors (PTSD, anxiety, function impairment), boys (PTSD and anxiety complaints), and younger children (pro-social behavior). However, there was an unintended harmful effect of intervention for girls on PTSD symptoms. | Moderators: gender (significant, boys more improvement in anxiety and PTSD), age (significant for pro-social behaviour, younger children more improvement), past exposure to violence, current war-related stressors (significant, children with lower levels improved more on PTSD and anxiety) Mediator: coping behaviour (not significant). | N/A | N/A |

*(Continued)*

**Table 2.** (Continued)

| Study | Country | Study type | Intervention (n participants) | Comparator (n participants) | Setting | Intervention location | Facilitators | Intervention duration | Age range | Participant gender (% female) | Inclusion criteria | Outcomes (scales) | Main results | Subgroup/moderation/mediation analyses | Qualitative evaluation | Theory of change |
|---|---|---|---|---|---|---|---|---|---|---|---|---|---|---|---|---|
| Tol et al., 2014 [67] | Burundi | Full RCT | CBI (n = 153) | Waitlist (n = 176) | Affected by conflict | School | Lay facilitators | 5 weeks | 8–17 | 50 | Children exposed to at least one potentially traumatic event, and who scored above the standard cut-off on symptom checklists for either PTSD, depression, or anxiety | Depression (DSRS); PTSD (CPSS) | No statistically significant differences were found. | Moderators: gender, age (significant - younger age increased hope only), displacement status (significant for hope and impairment only - intervention worsened these for children living in original villages), household size (significant - larger household better for PTSD and depression), family composition (significant - living with both parents better for PTSD and depression), exposure to traumatic events (significant - less trauma increased hope only), social capital (not significant). Mediators: coping, social support (not performed as not significant) | Focus group discussions, semi-structured interviews with children and care-givers, key informant interviews (content analysis). | N/A |
| Ugwu et al., 2022 [68] | Nigeria | Full RCT | REBT (n = 24) | Waitlist (n = 24) | Urban | Community centre | Psychologists | 12 weeks | 5–10 | 60 | Children with a learning disability (low score on WRAT) attending school regularly, with a high depresion score (CDI) | Depression (CDI) | REBT had a significant effect on the reduction of depression among children with learning disabilities. | N/A | N/A | N/A |

*(Continued)*

**Table 2.** (Continued)

| Study | Country | Study type | Intervention (n participants) | Comparator (n participants) | Setting | Intervention location | Facilitators | Intervention duration | Age range | Participant gender (% female) | Inclusion criteria | Outcomes (scales) | Main results | Subgroup/moderation/mediation analyses | Qualitative evaluation | Theory of change |
|---|---|---|---|---|---|---|---|---|---|---|---|---|---|---|---|---|
| Zafar et al., 2015 [69] | Pakistan | Full RCT | Didactic therapy (n=50) | Waitlist (n=50) | Urban | School | Psychologists | 8 weeks | 12–18 | 50 | Adolescents scoring >110 (DASS) | Anxiety (DASS anxiety subscale); Depression (DASS depression subscale) | There was significant improvement in the scores of male and female participants in the experimental groups, but not in the control groups. There was no significant difference between genders. | Moderator: gender (not significant) | N/A | N/A |

Legend: AADS = Asian Adolescent Depression Scale; APAI = Acholi Psychosocial Assessment Instrument; BAND = Belonging against Negative Thinking and Depression; BDI = Beck Depression Inventory; CBCL = Child Behaviour Checklist; CBI = Classroom Based Intervention; CBT = Cognitive Behavioural Therapy; CDI-S = Children's Depression Inventory-Short Version; CES-DC = Center for Epidemiological Studies Depression Scale for Children; CPDS = Child Psychosocial Distress Screener; CPSS = Child PTSD Symptom Scale; CRIES-13 = Children's Revised Impact of Event Scale; DASS = Depression, Anxiety and Stress Scale; DSRS = Depression Self Rating Scale; EASE = Early Adolescent Skills for Emotions; GAD-7 = Generalised Anxiety Disorder 7-item; HTQ = Harvard Trauma Questionnaire; IPT = Interpersonal Psychotherapy; KADS = Kutcher Adolescent Depression Scale; MBCT = Mindfulness-Based Cognitive Therapy; MFQ = Mood and Feelings Questionnaire; m-WET = modified Written Exposure Therapy; NGO = non-governmental organisation; PCL-5 = PTSD Checklist for DSM-5; PHQ-9 = Patient Health Questionnaire 9-item; PHQ-A = PHQ-9 modified for Adolescents; PSC-17 = Paediatric Symptom Checklist-17; PTSD-RI = UCLA Child/Adolescent PTSD Reaction Index; RADS-2 = Reynolds Adolescent Depression Scale, 2nd Edition; RCMAS = Revised Children's Manifest Anxiety Scale; REBT = Rational Emotive Behaviour Therapy; SCARED = Screen for Child Anxiety Related Disorders; TF-CBT = Trauma-Focused Cognitive Behavioural Therapy; TGIC = Traumatic Grief Inventory for Children; TRT = Teaching Recovery Techniques; WfR = Writing for Recovery; YRI = Youth Readiness Intervention.

**Table 3. Final programme theories organised under themes, with constituent CMOCs.**

| Theme | Programme theories | CMOCs |
|---|---|---|
| 1. Age | 1.1 When intervention content is matched to participants' level of cognitive development, children and adolescents will be better able to understand and implement learned skills, resulting in greater improvements in mental health outcomes. | (C) Younger girls with low literacy levels in Afghanistan…(M) wrote less in m-WET sessions and thus had less opportunity to engage with their experiences and emotions…(O) leading to less reduction in PTSD-associated distress [33].<br>(C) Use of m-WET with older adolescents in Afghanistan…(M) also acted as writing skill training and helped to motivate adolescents in learning a new skill while recovering…(O) leading to better participant engagement [33].<br>(C) A CBT-based intervention for depressed adolescents in Nigeria…(M) was deliberately weighted heavily towards behavioural rather than cognitive strategies, as these were…(O) easier for the adolescents to understand, and to explain [41].<br>(C) A trial of BAND in Thailand found that combining interpersonal and cognitive elements…(M) helped adolescents to develop a sense of belonging in early sessions, which equipped them with the communication skills and self-awareness to explore negative thoughts in later sessions…(O) resulting in enhanced ability to modify negative thinking [53].<br>(C) In a trial of Coping Cat for children aged 8–13 in Pakistan…(M) adaptation of language to simplify CBT principles helped children to understand the content involved…(O) resulting in improvements in anxiety symptoms [56].<br>(C) A CBT intervention for 16 year olds in Malaysia…(M) found that the comparable cognitive development level of the participants allowed them to understand the more complex cognitive component…(O) enabling them to target negative thought patterns [64]. |
| | 1.2 When interventions consist of age-appropriate activities, children and adolescents will be more engaged in sessions and will learn skills more effectively, resulting in greater improvements in mental health outcomes. | (C) When youth in Brazilian favelas were initially unsure about participating…(M) facilitators ensuring activities were fun and engaging effectively dealt with this uncertainty…(O) leading to better intervention attendance [40].<br>(C) In a trial of bibliotherapy for adolescent depression in the Philippines, younger participants…(M) who were avid readers found the intervention relevant and engaging…(O) resulting in greater benefit [51].<br>(C) In the BAND trial in Thailand, inclusion of age-specific activities and games at the start of each session…(M) motivated participants to arrive on time…(O) thus promoting attendance [53]. |
| 2. Gender | 2.1 When intervention content takes into account gender-specific issues in a given context, interventions will be more relevant to participants in targeting the right problems, resulting in greater improvements in mental health outcomes. | (C) In a TRT intervention in Brazilian favelas with ongoing drug-related violence, boys…(M) presented with more aggression than girls and required more validation of emotions and attention to non-retaliatory responses to conflict…(O) to be able to focus on the intervention [40].<br>(C) In a study of IPT in Uganda, girls…(M) were more comfortable talking about emotional problems in a group format than boys, who also had more substance misuse problems…(O) and thus boys benefited less from the intervention [44].<br>(C) In a trial of CBI in Nepal…(M) the active nature of the intervention seemed compatible with externalising expressions of distress common among boys…(O) resulting in improvements in aggression and behaviour-oriented difficulties [17]. |

*(Continued)*

**Table 3.** (Continued)

| Theme | Programme theories | CMOCs |
|---|---|---|
| 3. Social protective factors and stability | 3.1 In settings with greater environmental stability, participants will be better able to implement learned skills, leading to greater improvements in mental health outcomes. | (C) In the BAND trial in Thailand, stability of the home environment…(M) enabled adolescents to complete homework assignments wherein they applied skills learnt in the intervention to real-life contexts…(O) leading to reduction in depressive symptoms [53]. (C) In a trial of CBT for PTSD in Thailand, adolescents who continued homework activities for one month post-intervention…(M) were eventually able to consolidate and generalise learned skills, leading to…(O) significant improvements that were not immediately present at post-intervention [63]. |
| | 3.2 Children and adolescents that continue to be exposed to chronic stressors have less psychological reserves, and will therefore be less able to engage meaningfully with interventions and implement skills outside of direct instruction, leading to less improvement in mental health outcomes. | (C) Refugee youth in Jordan with ongoing stressors related to poverty, academic pressures, and prior traumatic events…(M) had limited capacity to experience reductions in distress…(O) leading to reduced remission across mental health outcomes [37]. (C) In an intervention for war-affected youth in Sierra Leone…(M) ongoing daily stressors and economic insecurity adversely affected…(O) intervention attendance [43]. (C) In a trial of CBI in Sri Lanka, children exposed to a higher number of ongoing war-related daily stressors…(M) were less able to make use of their learning in their daily lives…(O) showing fewer improvements in PTSD and anxiety [66]. |
| | 3.3 Children and adolescents with recourse to a higher level of social protective factors and support will usually be better able to engage with interventions and make use of learned skills, leading to greater improvements in mental health outcomes. | (C) In a trial of EASE in Jordan with a focus on parenting practice…(M) reduction of inconsistent disciplinary parenting behaviours…(O) led to greater improvements in internalising problems in children [45]. (C) In a trial of TRT in Lebanon, providing caregivers with a summary of the techniques taught to children in sessions…(M) enabled caregivers to support their children in practising techniques at home…(O) leading to improvements in trauma-related symptoms [32]. (C) In a trial of CBI in Burundi, children living with both parents…(M) had greater levels of social support and were better able to use the intervention to bolster resilience, (O) leading to greater improvements in depression and PTSD symptoms [67]. Conversely: (C) In an intervention among former child soldiers in Uganda, formerly abducted boys who experienced more stigma and social isolation compared to non-abducted boys…(M) found more meaningful opportunities to discuss their experiences in sessions and reintegrate into life, and made more proactive efforts…(O) leading to greater improvements in depression [42]. |
| 4. Cultural adaptation | 4.1 When interventions are culturally adapted in a 'deep' manner, they will more effectively match local conceptualisations of distress and appropriate responses, leading to greater improvements in mental health outcomes. | (C) A CBT intervention in Nigeria…(M) used religious-based coping self-talk, a commonly used technique in the local population, as well as local metaphors and analogies…(O) to teach participants how to apply cognitive techniques [41]. (C) A CBT intervention in Malaysia adapted interpersonal aspects…(M) to place greater emphasis on family and respect, which was perceived as more relevant and acceptable in a Malaysian cultural context…(O) leading to greater reductions in depression [64]. (C) A trial of CBI in Indonesia where the intervention was not specifically adapted for local conceptualisations of distress…(M) found that local predominantly somatic expressions of trauma were not effectively targeted, meaning that…(O) treatment effects for anxiety and depression were not found [23]. |

*(Continued)*

**Table 3.** (Continued)

| Theme | Programme theories | CMOCs |
|---|---|---|
| | 4.2 When interventions are culturally adapted in a 'deep' manner, they will be more acceptable to participants and work to reduce stigma, leading to greater improvements in mental health outcomes. | (C) In a trial of TF-CBT in Kenya and Tanzania…(M) psychological labels were avoided and sessions were referred to as "class" instead of "therapy"...(O) in order to make the intervention more acceptable and less stigmatising [47]. |
| | 4.3 When interventions are culturally adapted in a 'deep' manner, they will be better able to engage local stakeholders, leading to more effective intervention delivery and subsequently greater improvements in mental health outcomes. | (C) Cultural adaptation of the Coping Cat programme in Pakistan by using locally-relevant metaphors and imagery to explain concepts…(M) enhanced therapist satisfaction with the intervention and fidelity to the programme…(O) leading to greater focus on key skills [56]. |
| 5. Location of intervention delivery | 5.1 When interventions are delivered in non-clinical settings, they contribute to normalising difficulties and reducing stigma, leading to greater improvements in mental health outcomes. | (C) Delivering group interventions in school…(M) can normalise difficulties and promote peer support through which problem-solving and coping skills can be practised in the safety of a group setting… (O) leading to reduction of stigma as well as opportunities to consolidate learned skills [22]. (C) Delivering a group CBT intervention in Chinese schools…(M) circumvented stigma as a treatment barrier…(O) improving accessibility [58]. |
| | 5.2 When interventions are delivered in existing institutions, they are able to build on existing networks of group and peer support, resulting in greater improvements in wellbeing and mental health outcomes. | (C) The shared experience of group work…(M) can promote participants to form smaller support groups to practise techniques together outside of formal sessions…(O) accelerating recovery by simultaneously providing peer support and improving skills [60]. (C) School-based protective factors in place before an intervention begins…(M) may contribute to development of resilience across a range of domains…(O) enabling generalisation of gains beyond the programme [38]. |
| | 5.3 When interventions are delivered within existing institutions, participants can be more easily identified and recruited, meaning that the intervention targets those who need it the most, leading to greater improvements in mental health outcomes. | (C) A CBT intervention set in a Nigerian school…(M) allowed a large proportion of young people to be reached easily without extra logistics…(O) meeting a recognised need in school settings [41]. |
| | 5.4 When interventions are embedded in established community institutions, key stakeholders can be more easily empowered, resulting in greater acceptance of and engagement with interventions, leading to greater improvements in mental health outcomes. | (C) Delivering a mind-body skills intervention in schools in Kosovo… (M) involved teachers in activities and practising techniques…(O) enhancing the social support they provided to their students [50]. (C) In a trial of guided self-help in Pakistan, when school principals were approached via established contacts…(M) they took the project seriously and perceived a sense of trust and security…(O) resulting in better intervention uptake [55]. |
| | 5.5 When interventions are delivered in schools, dissemination of learning and skills in a variety of forums beyond the intervention group can lead to school-wide improvements in mental health outcomes. | (C) In a trial of a trauma-focused group treatment in Bosnia…(M) students in the intervention group shared the skills they learned with classmates, and counsellors disseminated coping skills in posters and at conferences…(O) leading to improvements in mental health outcomes in the untreated comparison group [57]. |

to externalising expressions of distress among boys compared to girls being more comfortable talking about emotional problems in groups. We synthesised the CMOCs into programme theory 2.1, recognising that gender-specific issues vary across contexts and therefore cannot be generalised, but in all cases require careful matching of intervention content. Moderation analyses identified gender as significant in five RCTs [17,44].

### Theme 3 - Social protective factors and stability

We identified important roles for environmental stability (programme theory 3.1), chronic stressors and psychological reserves (programme theory 3.2), and levels of social protective factors (programme theory 3.3). We formulated nine

CMOCs. Longer-term follow-up data emphasised the importance of participants consolidating and generalising learned skills, which is easier in stable settings compared to situations with ongoing conflict or instability in the home environment; programme theory 3.1 [63]. Process evaluation data highlighted the role of ongoing stressors, such as trauma exposure, in limiting capacity to experience Reductions in distress [37]; programme theory 3.2, supported by significant results in a moderation analysis for a different RCT [66]. The role of parents and parenting practices was a key interacting mechanism in two mediation analyses [37,45], forming the basis of programme theory 3.3. This was refuted by another CMOC supported by moderation analyses of an RCT in Uganda, which hypothesised that formerly abducted boys who experienced more stigma and social isolation found more meaningful opportunities and made proactive efforts to discuss their experiences and reintegrate into social life, leading to greater improvements in depression [42]. Through refutational synthesis, we further qualified programme theory 3.3, suggesting that extreme levels of instability might create a particularly urgent impetus.

### Theme 4 - Cultural adaptation

Programme theories centred on matching local conceptualisations of distress (programme theory 4.1), reducing stigma (programme theory 4.2), and engaging local stakeholders (programme theory 4.3). We formulated five CMOCs. Qualitative data from focus groups and expert panels in the included reports, [41] consistent with existing literature, led us to differentiate 'superficial' from 'deep' cultural adaptation (going beyond simply translating language to considering metaphors, content, concepts, goals, and methods, and their applicability to a unique cultural context) [70]. We compared the effectiveness of interventions with deep adaptation to the lack of effect of an intervention that failed to match local, predominantly somatic, conceptualisations of distress; programme theory 4.1 [23]. We developed programme theories 4.2 and 4.3 by considering the applicability of the remaining CMOCs on a more abstract level. There were no relevant moderation or mediation analyses.

### Theme 5 - Location of intervention delivery

As the majority of RCTs were conducted in schools, our programme theories predominantly relate to mechanisms particularly pertinent to this setting. Nonetheless, other non-clinical and established community settings can also be conducive for the activation of mechanisms. Key mechanisms included normalising difficulties and reducing stigma (programme theory 5.1), building on existing networks of group and peer support (programme theory 5.2), identification and recruitment of participants (programme theory 5.3), empowerment of stakeholders (programme theory 5.4), and dissemination of learning (programme theory 5.5). We formulated eight CMOCs building on participant and facilitator feedback from focus groups. We expected that public and private schools might have different levels of scope for activating protective factors; however moderation analyses of school type in two RCTs failed to find significant results [38,39]. Consequently, when synthesising CMOCs into programme theories we focused on broader networks of support rather than individual school-based protective factors (programme theory 5.2).

One RCT suggested a theme related to lay-facilitator engagement, noting key factors promoting facilitator engagement: working with a reputable organisation, co-working with another facilitator to create a collegial atmosphere, and being given an opportunity to develop skills. These were hypothesised to promote fidelity to the intervention model, leading to improved outcomes [38]. However, data were insufficient to develop this theme.

### Risk of bias

We rated nineteen RCTs 'low risk'; 15 'some concerns'; and 5 'high risk' of bias (S1 Table). The most common source of risk of bias was missing outcome data. Risk of bias had no significant effect on results in the meta-analysis.

### Effect of psychosocial interventions

Depression symptoms: the pooled effect size was -0.72 (95% CI -1.01 to -0.42, p<0.001), with high heterogeneity (I-squared=0.89, 95% CI 0.85 to 0.92); see Forest plot (Fig 2). In univariate regressions, only age and exposure to conflict

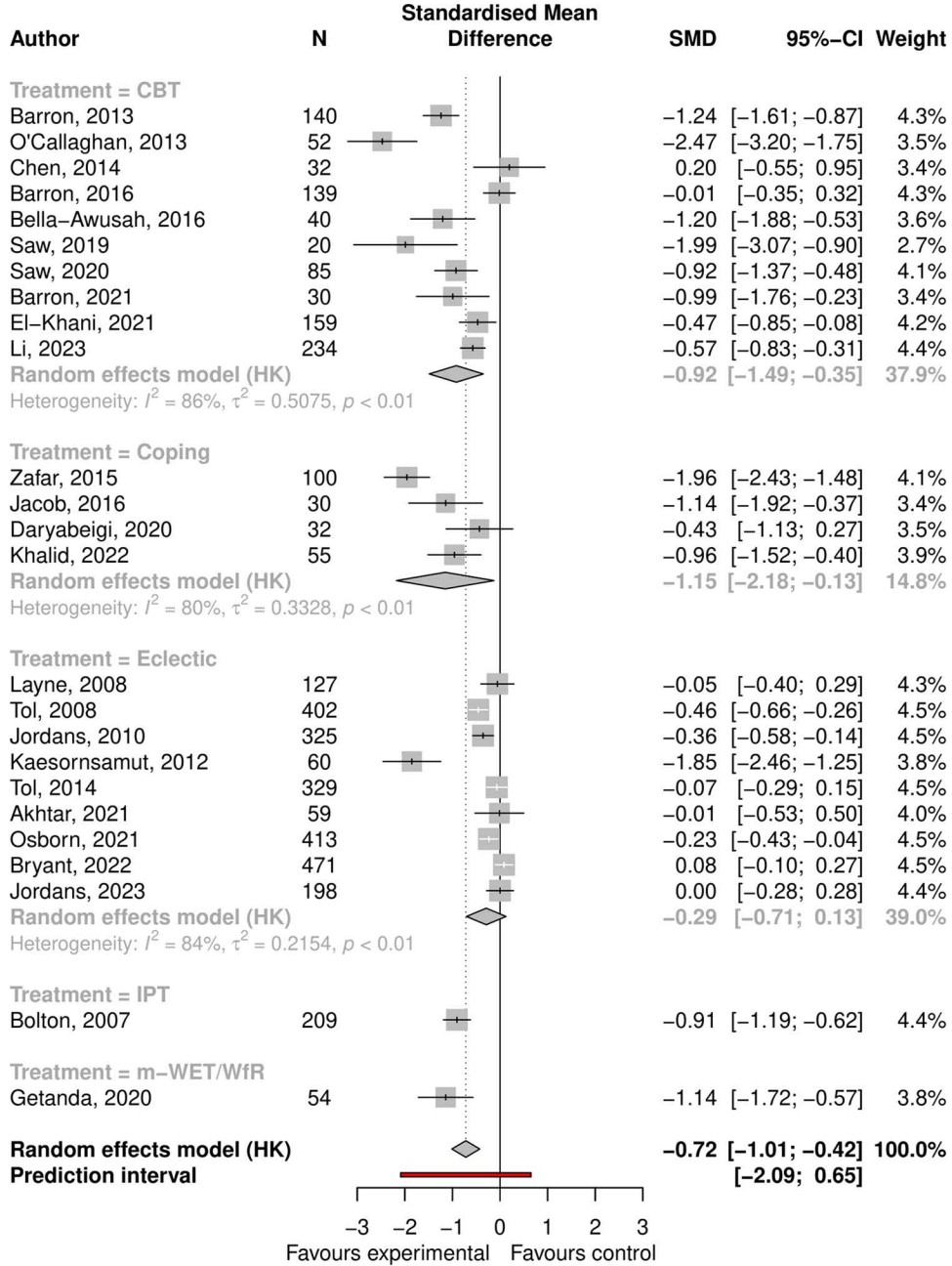

**Fig 2. Forest plot showing effect of psychosocial interventions for depression symptoms.**

showed significant associations with ES. ES were larger for older participants versus younger (difference = 0.6, 95% CI -1.14 to -0.057, p = 0.032) and in non-conflict-affected populations versus conflict-affected populations (difference = 0.6, 95% CI 0.05 to 1.15, p = 0.033). The Funnel plot showed clear asymmetry (S1 File), and Egger's test was significant at p < 0.001 confirming small study effects. Following adjustment using limit meta-analysis the pooled effect was no longer significant (-0.31, 95% CI -0.71 to 0.10, p = 0.137).

Anxiety symptoms: the pooled effect size was -0.90 (95% CI -1.57 to -0.23, p=0.014), with high heterogeneity (I-squared=0.93, 95% CI 0.89 to 0.95); see Forest plot (Fig 3). In univariate regressions no study characteristics were associated with ES. The Funnel plot showed some asymmetry (S1 File), and Egger's test was significant at p=0.022 confirming small study effects. Using limit meta-analysis to adjust for small study effects reduced the overall ES to -0.3.

PTSD symptoms: the pooled effect size was -0.71 (95% CI -1.05 to -0.38, p<0.001), with high heterogeneity (I-squared=0.91, 95% CI 0.88 to 0.93); see Forest plot (Fig 4). In univariate regressions only intervention type was significantly associated with effect size (p=0.006). NET/WET-based interventions had the greatest effect sizes, with eclectic the smallest. The Funnel plot showed clear asymmetry (S1 File), and Egger's test was significant at p=0.002 confirming small study effects. Adjusting ES for small study effects using limit meta-analysis meant the overall pooled effect was no longer significant (0.11, 95% CI -0.58 to 0.36, p=0.653).

In sensitivity analyses to remove outliers, heterogeneity (I-squared) and effect sizes decreased but remained significant (p<0.05) across depression, anxiety, and PTSD outcomes (S2 File).

Among our programme theories, only themes one (age) and three (social protective factors and stability) yielded significant results in univariate regressions. For depression symptoms, ES were larger for older children and in populations without exposure to conflict. Full meta-regression results are available in S3 File.

## Discussion

This is, to our knowledge, the first realist systematic review of group psychosocial interventions for depression, anxiety, and PTSD among children and adolescents in LMICs. Previous reviews have found heterogeneous results; it is clear that

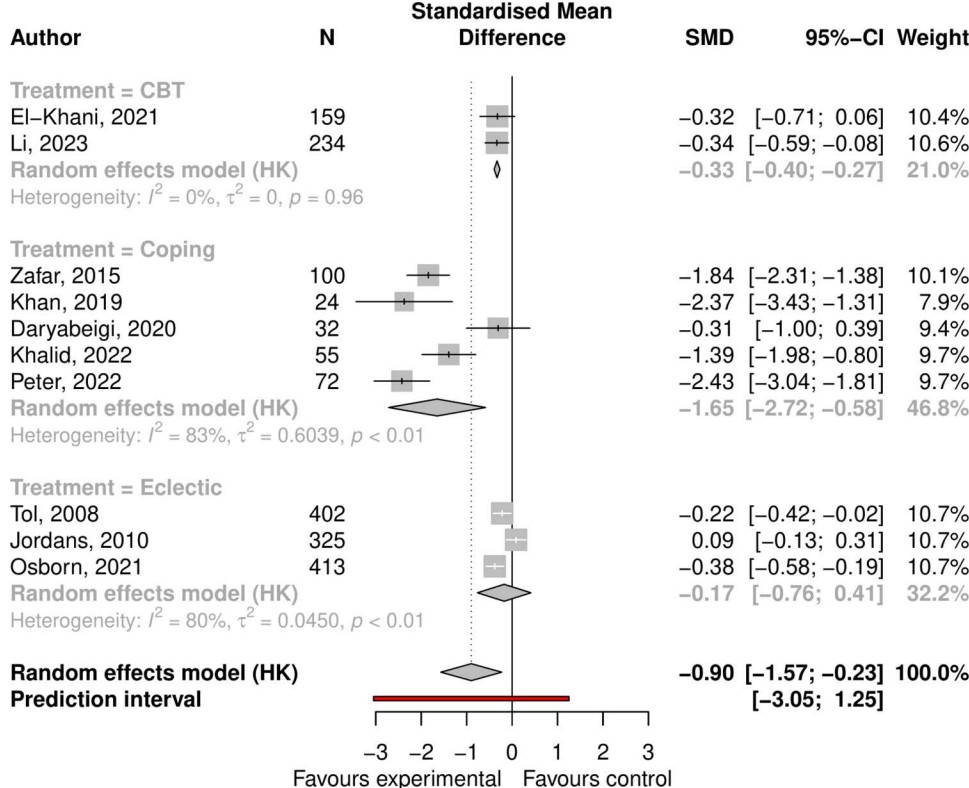

**Fig 3. Forest plot showing effect of psychosocial interventions for anxiety symptoms.**

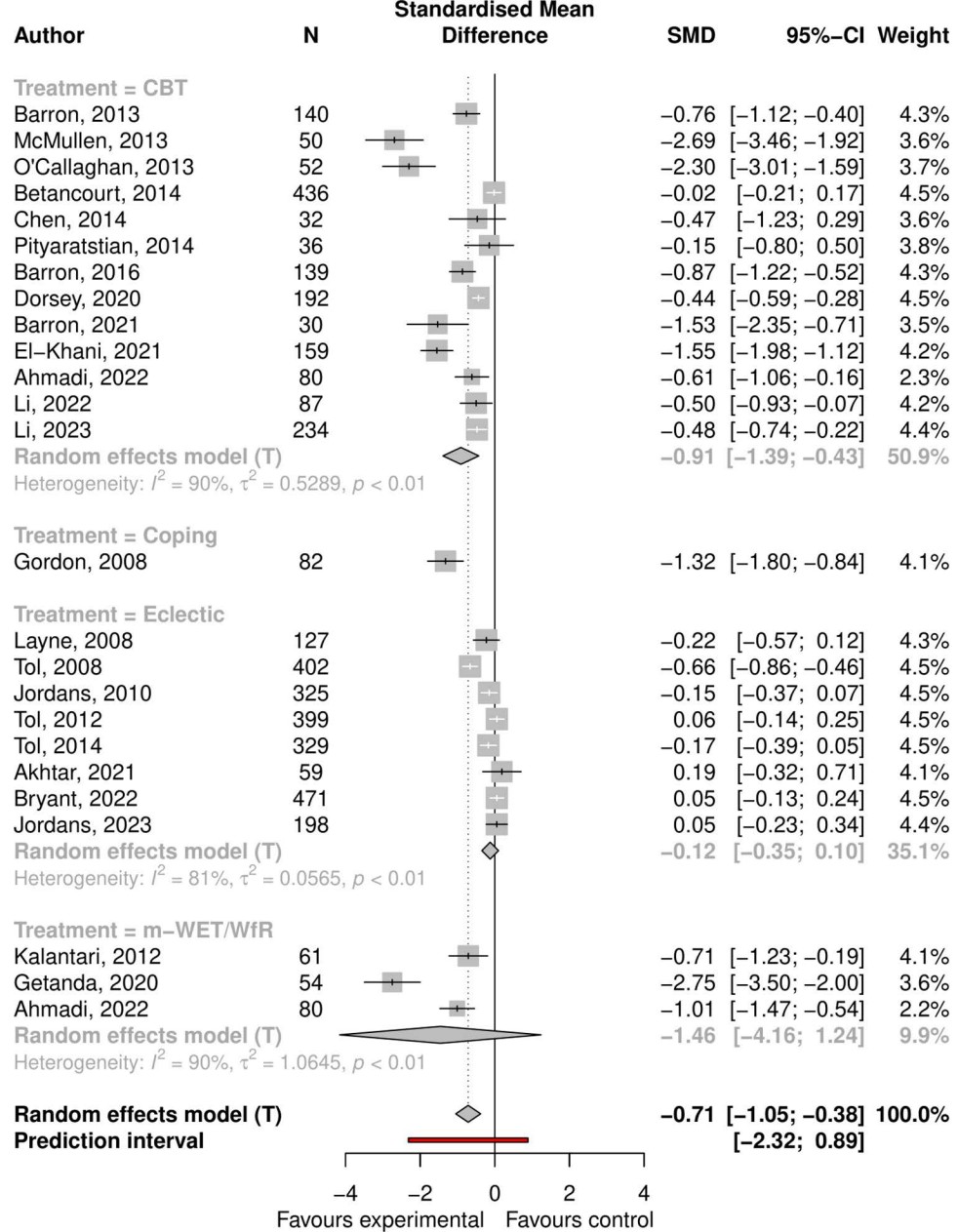

**Fig 4. Forest plot showing effect of psychosocial interventions for PTSD symptoms.**

psychosocial interventions can be highly effective, but can also have very little impact [20,71,72]. Insufficient attention has been paid to the interaction of contextual factors and mechanisms that explain some aspects of this heterogeneity [73]; this review has sought to make initial suggestions and inform further research.

Overall results revealed a large effect of psychosocial interventions, with substantial heterogeneity. Our meta-analysis found slightly larger pooled effect sizes and similar heterogeneity compared to a recent meta-analysis exploring the same outcomes [74], and slightly smaller pooled effects compared to one with a narrower focus on psychotherapy interventions

[9]. Treatment type was associated with the size of effects on PTSD symptoms, with narrative/written-exposure-therapy-based interventions having the greatest effect and CBI the least. The overlap between outcomes reflects both rates of comorbidity and non-specificity of diagnostic criteria, rating scales, and screening instruments. Previous reviews have argued for transdiagnostic interventions having wider applicability and feasibility in addressing a range of common problems rather than disorder-specific criteria [25,75].

We identified programme theories and constituent CMOCs that hypothesise key aspects of the success or failure of interventions, summarised under five themes of interest. Children and adolescents of different ages respond differently to intervention content. Interventions should be designed to match the developmental level of the target population. Children and younger adolescents benefit most from behavioural interventions that are practical and focus on active skills, whereas older adolescents may be better able to understand and apply more cognitive interventions. This is consistent with trajectories of development moving towards a greater capacity for metacognitive engagement towards later adolescence. In certain contexts where the cultural milieu may place less emphasis on introspective analysis of cognitive patterns, however, such a linear trajectory may not apply [76,77]. These differences may help to explain some of the heterogeneity of findings across the included studies. If interventions with a heavy cognitive focus are to be used, one approach could be to combine behavioural and cognitive elements into a single intervention, beginning with behavioural emphases to promote engagement and initial gains, which then allows effective socialisation to the cognitive model [53]. Interventions should involve age-appropriate, enjoyable activities, such that participants will attend, engage, and work to apply learned skills in real-life contexts.

Gender can affect intervention relevance. Different activities for boys and girls might promote better engagement. Different skills might need to be emphasised to target particular behavioural or cognitive patterns that feature in local gender norms, and gender-specific problems might need to be considered [40,44]. Gender may also affect participation in group therapy [78]. Effective interventions might therefore look different for gender-segregated groups, or incorporate gender-specific components in mixed groups [77]. Little is known about how this relates to gender diverse and non-binary gender categorisations, especially in LMICs. None of the included studies in our review incorporated data for gender diverse participants, and more research is needed.

Interventions might work best when children and adolescents have a higher level of social protective factors (including financial security and social status) and environmental stability that allows them to engage with therapy, implement learned skills outside of sessions, and receive support to continue making changes in the post-intervention period [19,23,79]. Special attention may be needed to enable children and adolescents with ongoing stress and trauma to derive benefit. Notably, there was a discrepancy in programme theory 3.3 (children and adolescents with recourse to a higher level of social protective factors and support will be better able to engage with interventions and make use of learned skills); a refuting CMOC suggested that it was those boys that experienced the highest levels of stigma and isolation that benefited most. Possibly this could be related to the unique context of the study in a displaced-persons camp with formerly-abducted child soldiers, or because these boys had higher baseline levels of symptoms and hence greater room for improvement [42]. Perhaps extreme levels of instability and trauma history, alongside stigma associated with abduction, compelled even the most traumatised adolescents to engage and benefit. Meta-regressions showed that for depression, interventions appeared to be less effective for conflict-affected populations. The number of studies was too small to provide definitive conclusions, and more research is needed to refine hypotheses about the relevance of ongoing conflict and trauma exposure. More work is needed to better understand the potential relevance of factors such as uneven economic development, rural-urban differences, adversities such as bereavement, and ongoing trauma exposure, in order to allow for more precise insights.

Culturally adapted interventions should go beyond simple translation of language to include culturally relevant metaphors, content, concepts, goals, and methods [70]. Doing so promotes engagement, reduces stigma, and considers local conceptualisations of distress and appropriate responses to these. Otherwise, content may be less relevant to

 

participants' problems. However, our CMOCs related to this theme were formulated primarily from qualitative data and we were unable to test programme theories statistically; our findings contrast with a meta-analysis that found locally adapted interventions to be less effective than direct imports [9]. One potential explanation for this inconsistency is that adaptations to interventions are rarely described in detail by authors and may differ greatly in their nature and effectiveness. It might be that deep adaptations are more effective than superficial adaptations but have been less used in past studies. Better reporting is needed to facilitate evaluation of different kinds of adaptation and to identify appropriate methodology to adapt interventions, ensuring adaptations optimise rather than worsen outcomes.

The school setting may provide unique advantages. This is consistent with findings from a systematic review of school-based interventions in LMICs showing overall effectiveness for mental health conditions; and a realist review of universal school-based interventions in humanitarian contexts for non-clinical wellbeing [80,81]. In schools, participants can be easily identified. Difficulties can be normalised, potentially reducing stigma, and teachers and principals can be empowered as stakeholders. Existing peer support networks and supportive group dynamics can be built upon, creating an environment conducive to therapeutic work that is rich in curative factors [82]. Networks of information sharing can allow those that are not actively participating in an intervention to benefit indirectly. However, this depends on school climate; schools can also be hostile environments with children subject to bullying and corporal punishment, as well as worsening of stigma. Implementing lay-facilitator delivered interventions in under-resourced school settings can be challenging however, and issues including facilitator training, supervision, and sustainability warrant further attention.

Limitations of the review relate to the included RCTs and the review methods. Only 19 of the 38 included RCTs were deemed low risk of bias. Although this is a common issue in RCTs testing complex psychosocial interventions, the risk of bias potentially undermines the robustness of pooled effect sizes. Effect sizes were not significantly affected after adjusting for overall risk of bias, but did become non-significant after accounting for small study samples in the limit meta-analysis. There is therefore a need for cautious interpretation of our findings as well as a need for more high-quality RCTs to be conducted. Few RCTs explored differential outcomes for subgroups, possibly because they were underpowered. Consequently, we were unable to quantitatively test all our CMOCs through meta-regressions. Other potentially important mediators, such as facilitator training, session dose, frequency, and intervention duration were not able to be tested and could be further investigated. The meta-regressions done at the study level, without individual participant data, are prone to the ecological fallacy and ignore randomisation, and are thus limited in how severely they can test CMOCs. Individual participant data meta-analyses might be helpful, although data on important contextual factors may not have been collected. Therapy manuals might set out theorised mechanisms, but this may not match what happens in reality, and few RCTs were set up to explicitly test mechanisms. The large amount of comorbidity and diagnostic overlap in the included RCTs made it difficult to separate out what mental health conditions were being treated beyond non-specific psychological distress. Screening tools and rating scales tend to prioritise sensitivity over specificity and therefore trials tend to include a heterogenous mixture of psychopathology. However, the universality of psychiatric diagnoses across cultures and contexts is debatable, and perhaps too zealous an emphasis on strict diagnostic criteria risks a mismatch with local conceptualisations of distress.

Key strengths of this review include a systematic search strategy and the integration of realist analysis with quantitative methods. We took a different approach to that proposed by Bonell et al.; [11] we used an initial scoping search to structure our data extraction and formulation of CMOCs and we did not synthesise process evaluation data, because this was lacking in the included reports.

Our findings call for further research into psychosocial interventions for children and adolescents in LMICs. There is an urgent need to rigorously and systematically test theory by setting out targeted CMOCs explicitly. More attention to mediators, moderators, and subgroup analyses can help to delineate mechanisms and identify important contextual factors, which might inform new theory. Each of the themes identified in this review merits further exploration and could serve as a starting point for realist evaluations. Although we know that schools can be a good location for intervention

delivery, variation between schools needs exploration, with attention to the factors, including school climate, that make interventions particularly successful in one school over another. This might help better match interventions, as well as suggest changes that could make the school environment more conducive. Inconsistencies within the theme 'social protective factors and stability' also require further exploration. Qualitative research is called for, to better understand the mechanisms and processes through which different psychosocial interventions bring about change, and how these mechanisms might interact on a deeper level in particular intervention contexts. This includes attention to 'deep' cultural adaptation and exploring what this really means for a mental health intervention.

Our findings, in conjunction with what is already known, can be of help to policymakers and researchers alike. The paucity of resources in many LMIC settings, as well as the urgent unmet clinical need with its potential for wide ranging and lasting implications, means that any intervention chosen must be a good fit for the context and target population, and we hope that our suggestions can contribute towards this.

## Supporting information

**S1 Appendix. Systematic review search strategy.**
(PDF)

**S1 Text. List of studies excluded at full-text screening, with reasons.**
(PDF)

**S1 Table. Risk of bias ratings: Cochrane risk of bias 2 tool.**
(PDF)

**S2 Table. Full data extracted from studies used for meta-analysis.**
(XLSX)

**S1 File. Funnel plots.**
(PDF)

**S2 File. Sensitivity analyses.**
(PDF)

**S3 File. Meta-regressions.**
(PDF)

**S4 File. PRISMA checklist.** From: Page MJ, McKenzie JE, Bossuyt PM, Boutron I, Hoffmann TC, Mulrow CD, et al. The PRISMA 2020 statement: an updated guideline for reporting systematic reviews. BMJ 2021;372:n71. https://doi.org/10.1136/bmj.n71
(PDF)

## Author contributions

**Conceptualization:** Tahir Jokinen, Kelly Rose-Clarke.

**Data curation:** Tahir Jokinen.

**Formal analysis:** Tahir Jokinen, Mujahed Abassi, John Hodsoll, Katie H Atmore, Kelly Rose-Clarke.

**Investigation:** Tahir Jokinen, Mujahed Abassi, John Hodsoll, Katie H Atmore.

**Methodology:** Tahir Jokinen, John Hodsoll, Katie H Atmore, Chris Bonell.

**Project administration:** Tahir Jokinen.

**Software:** John Hodsoll, Kelly Rose-Clarke.

**Supervision:** Chris Bonell, Kelly Rose-Clarke.

**Validation:** Tahir Jokinen, Mujahed Abassi.

**Writing – original draft:** Tahir Jokinen, John Hodsoll.

**Writing – review & editing:** Tahir Jokinen, Mujahed Abassi, John Hodsoll, Katie H Atmore, Chris Bonell, Kelly Rose-Clarke.

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
