## [Decision Letter · Decision Letter 0]

20 Oct 2025

PMEN-D-25-00129

Group psychosocial interventions for anxiety, depression, and post-traumatic stress disorder in children and adolescents in low- and middle-income countries: a realist systematic review and meta-analysis of randomised controlled trials

PLOS Mental Health

Dear Dr. Kelly Rose-Clarke,

Thank you for submitting your manuscript to PLOS Mental Health. After careful consideration, we feel that it has merit but does not fully meet PLOS Mental Health’s publication criteria as it currently stands. Therefore, we invite you to submit a revised version of the manuscript that addresses the points raised during the review process.

We look forward to receiving your revised manuscript.

Kind regards,

Olumide Thomas Adeleke, MBBS, FWACP

Academic Editor

PLOS Mental Health

Journal Requirements:

1. In the online submission form, you indicated that “The datasets used and/or analysed during the current study are available from the corresponding author on reasonable request.”.

a) In a public repository,

b) Within the manuscript itself, or

c) Uploaded as supplementary information.

2. Please provide separate main figure files in .tif or .eps format only and ensure that all files are under our size limit of 10MB.

For more information about how to convert your figure files please see our guidelines: https://journals.plos.org/mentalhealth/s/figures

3. Main tables (Tables 1, 2 and 3) should not be uploaded as individual files. Please remove these files and include the Tables in your manuscript file as editable, cell-based objects. For more information about how to format tables, see our guidelines: https://journals.plos.org/globalpublichealth/s/tables

4. As required by our policy on Data Availability, please ensure your manuscript or supplementary information includes the following:

5. We have noticed that you have uploaded Supporting Information files, but you have not included a list of legends. Please add a full list of legends for your Supporting Information files before or after the references list.

Additional Editor Comments (if provided):

Reviewers' comments:

Reviewer's Responses to Questions

**Comments to the Author**

1. Does this manuscript meet PLOS Mental Health’s publication criteria?

Reviewer #1: Yes

Reviewer #2: Yes

2. Has the statistical analysis been performed appropriately and rigorously?

Reviewer #1: Yes

Reviewer #2: Yes

3. Have the authors made all data underlying the findings in their manuscript fully available (please refer to the Data Availability Statement at the start of the manuscript PDF file)?

Reviewer #1: Yes

Reviewer #2: Yes

4. Is the manuscript presented in an intelligible fashion and written in standard English?

Reviewer #1: Yes

Reviewer #2: Yes

Reviewer #1: This manuscript presents a rigorous and methodologically innovative realist systematic review combined with meta-analysis, addressing a critical and underexplored question of how and why psychosocial interventions work for youth in low- and middle-income countries. The integration of realist synthesis principles with conventional systematic review and meta-analytic techniques represents a major methodological contribution and aligns well with global mental health priorities. The emphasis on context-mechanism-outcome configurations and program theories offers valuable explanatory insights beyond what standard effect-size estimates can provide. Here are a few suggestions that authors can consider to further strengthen the manuscript:

1. The inclusion of studies spanning early childhood to late adolescence introduces developmental variability that likely influences intervention responsiveness. Consider explicitly categorizing age groups (e.g., early childhood, early adolescence, late adolescence) in the meta-regression analyses. The finding that older adolescents experienced greater reductions in depression may reflect their greater capacity for metacognitive engagement, whereas younger children may respond better to behaviorally oriented approaches. A brief developmental framing would enhance the interpretation of differential effects.

2. The analysis does not account for gender-diverse, an increasingly important demographic in global mental health. While data may be limited, acknowledging this gap explicitly (perhaps in the limitations) would signal inclusiveness and transparency.

3. The manuscript notes the role of cultural adaptation in reducing stigma but would benefit from concrete illustrations or case examples (e.g., adaptations to local language, idioms of distress, or community delivery formats). This would strengthen the explanatory power of relevant CMOCs.

4. Given the realist aims, it is somewhat surprising that qualitative studies and grey literature were excluded. A brief rationale would clarify this decision and address potential reader concerns about omitted evidence.

5. Only 19 of the 38 included RCTs were deemed low risk of bias. Consider discussing the implications of this in more detail, particularly regarding the robustness of pooled effect sizes. The fact that adjusted effect sizes became non-significant after accounting for bias further underscores the need for cautious interpretation.

6. A few covariates yielded non-significant results. These null findings could be contextualized further. For example, were these variables theoretically expected to influence outcomes? A more detailed discussion might help clarify their role or lack thereof.

7. Where intervention effects were non-significant, consider hypothesizing potential reasons, such as ongoing trauma exposure, contextual stressors, or inadequate intervention intensity. This might also apply to the finding that CBT outperformed other modalities in some outcomes, despite sparse comparative trials.

8. Since analyses are conducted at the study level, individual-level mechanisms cannot be inferred with confidence. For example, it is unclear whether older adolescents benefited more as individuals, or whether studies targeting them were better designed. Perhaps good to acknowledge the limitations of ecological fallacy and recommend future individual participant data meta-analyses to deepen this line of inquiry?

9. The manuscript could briefly address the feasibility of implementing lay-delivered interventions in under-resourced school settings. Issues like training, supervision, and sustainability warrant mention, especially as this delivery model is central to many included studies.

10. Consider acknowledging other potentially important moderators that were not assessed, such as facilitator training, session dose, frequency, and intervention duration and perhaps recommend them as areas for future investigation?

11. The covariate analysis partially advances the realist objectives by empirically exploring heterogeneity. However, linking the findings more explicitly back to the CMOCs and program theories in the discussion would improve narrative coherence. Highlighting how significant and null results map onto or challenge program theories would enrich interpretive clarity.

Reviewer #2: well done research on group psychosocial interventions for anxiety, depression, and post-traumatic stress

disorder in children and adolescents in low- and middle-income countries: a realist

systematic review and meta-analysis of randomised controlled trials.

**Do you want your identity to be public for this peer review?** For information about this choice, including consent withdrawal, please see our Privacy Policy

Reviewer #1: **Yes:**  Sachin Shinde

Reviewer #2: No

---

## [Editor Report · Decision Letter 1]

17 Dec 2025

Group psychosocial interventions for anxiety, depression, and post-traumatic stress disorder in children and adolescents in low- and middle-income countries: a realist systematic review and meta-analysis of randomised controlled trials

PMEN-D-25-00129R1

Dear Dr Rose-Clarke,

We are pleased to inform you that your manuscript 'Group psychosocial interventions for anxiety, depression, and post-traumatic stress disorder in children and adolescents in low- and middle-income countries: a realist systematic review and meta-analysis of randomised controlled trials' has been provisionally accepted for publication in PLOS Mental Health.

Best regards,

Olumide Thomas Adeleke, MBBS, FWACP

Academic Editor

PLOS Mental Health